# COMPOSITIONAL VISUAL GENERATION WITH ENERGY BASED MODELS

## ABSTRACT

Humans are able to both learn quickly and rapidly adapt their knowledge. One major component is the ability to incrementally combine many simple concepts to accelerates the learning process. We show that energy based models are a promising class of models towards exhibiting these properties by directly combining probability distributions. This allows us to combine an arbitrary number of different distributions in a globally coherent manner. We show this compositionality property allows us to define three basic operators, logical conjunction, disjunction, and negation, on different concepts to generate plausible naturalistic images. Furthermore, by applying these abilities, we show that we are able to extrapolate concept combinations, continually combine previously learned concepts, and infer concept properties in a compositional manner.

## 1 INTRODUCTION

Humans are able to rapidly learn new concepts and continually integrate them among their prior knowledge. The key ingredient in enabling this is the ability to compose increasingly complex concepts out of simpler ones, and recombining and reusing concepts in novel ways (Fodor & Lepore, 2002). By combining a finite number of primitive components, humans can create an exponential number of new concepts, and use them to rapidly explain current and past experiences (Lake et al., 2017). We are interested in enabling such compositionality capabilities in machine learning systems, particularly in the generative modeling context.

Past efforts in machine learning to incorporate compositionality have attempted it in several distinct ways. One has been to decompose data into disentangled factors of variation and situate each datapoint in the resulting - typically continuous - factor vector space (Vedantam et al., 2018; Higgins et al., 2018). The factors can either be explicitly provided or learned in an unsupervised manner. In both cases, however, the dimensionality of the factor vector space is fixed and defined prior to training. This makes it difficult to introduce new factors of variation, which may be necessary to explain new data, or to differently taxonomize past data. Another approach to incorporate the compositionality is to spatially decompose an image into a collection of objects, each object slot occupying some pixels of the image defined by a segmentation mask (van Steenkiste et al., 2018; Greff et al., 2019). Such approaches can generate visual scenes with multiple objects, but may have difficulty in generating interacting effects between objects. These two incorporations of compositionality are typically seen as distinct, with very different underlying implementations.

In this work, we propose to implement compositionality ideas via energy based models (EBMs). Instead of an explicit vector of factors that is input to a generator function, or object slots that are blended to form an image, our unified treatment defines factors of variation and object slots via energy functions. Each factor is represented by an individual scalar energy function that takes as input an image and outputs a low energy value if the factor is exhibited in the image. Images that exhibit the factor can then be generated implicitly as a result of an MCMC process that minimizes the energy. Importantly, it is also possible to run MCMC process on some *combination* of energy functions to generate images that exhibit multiple factors or multiple objects, in a globally coherent manner.

There are several ways to combine energy functions. One can add or multiply distributions defined by the energy functions (as in mixtures (Shazeer et al., 2017; Greff et al., 2019) or products (Hinton, 2002) of experts). We view these as probabilistic instances of logical operators over concepts. Instead of using one, we consider three operators: logical conjunction, disjunction, and negation (illustrated in Figure 1). We can then flexibly and recursively combine multiple energy functions via these operators. More complex operators (such as implication) can be formed out of our base operators.

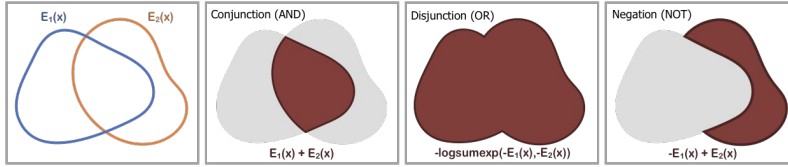

Figure 1: Illustration of logical composition operators over energy functions $E_1$ and $E_2$ (drawn as level sets).

EBMs with such logical composition operators enable several capabilities. They allow defining new concepts (factors) implicitly via examples. This is similar to learning to generate images in a few-shot setting (Reed et al., 2017), with the distinction that instead of learning to generate holistic images from few examples, we learn *properties* from examples in a way that can then be flexibly combined with other previously learned concepts. This allows new concepts to be added on demand in a continual manner by simply learning a new energy function from examples, and which again can be combined with all past concepts. Additionally, finely controllable image generation can be enabled by specifying the desired image via a collection of logical clauses, with applications to neural scene rendering (Eslami et al., 2018).

Our contributions are as follows: first, while composition of energy-based models has been proposed in abstract settings before (Hinton, 2002), we show that it can be used to generate plausible natural images. Second, we propose to combine energy models based on logical operators which can be chained recursively, allowing controllable generation based on a collection of logical clauses. Third, we demonstrate unique advantages of such an approach, such as extrapolation to concept combinations, continual addition of new energy functions, and ability to infer concept properties.

## 2 METHOD

In this section, we first give a background overview of EBMs and then define three different basic logic operators on them. The components of these operators can be learnt independently and incrementally combined to support continual learning. Furthermore, the operators themselves can be combined to support nested compositions.

### 2.1 ENERGY BASED MODELS

EBMs represent data by learning an unnormalized probability distribution across the data. For each data point $\mathbf{x}$, an energy function $E_\theta(\mathbf{x})$, parameterized by a neural network, outputs a scalar real energy such that

$$p_\theta(x) \propto e^{-E_\theta(x)}. \tag{1}$$

To train an EBM on a data distribution $p_D$, we follow the methodology defined in (Du & Mordatch, 2019), where a Monte Carlo estimate (Equation 2) of maximum likelihood is minimized.

$$\nabla_\theta \mathcal{L} = \mathbb{E}_{x^+ \sim p_D} E_\theta(x^+) - \mathbb{E}_{x^- \sim p_\theta} E_\theta(x^-). \tag{2}$$

To sample $x^-$ from $p_\theta$ for both training and generation, we use MCMC based off Langevin dynamics (Welling & Teh, 2011). Samples are initialized from uniform random noise and are iteratively refined following Equation 3

$$\tilde{\mathbf{x}}^k = \tilde{\mathbf{x}}^{k-1} - \frac{\lambda}{2} \nabla_{\mathbf{x}} E_\theta(\tilde{\mathbf{x}}^{k-1}) + \omega^k, \ \omega^k \sim \mathcal{N}(0, \lambda), \tag{3}$$

where $k$ is the $k^{th}$ iteration step and $\lambda$ is the step size. We refer to each iteration of Langevin dynamics as a negative sampling step. We note that this form of sampling allows us to generate samples from distributions composed of $p_\theta$ and other distributions by using the gradient of the modified distribution. We use this ability to generate from multiple distributions that allow various different forms of compositionality that we detail below.

### 2.2 COMPOSITION OF ENERGY-BASED MODELS

We next present different ways that EBMs can compose. We consider a set of independently trained EBMs, $E(\mathbf{x}|c_1), E(\mathbf{x}|c_2), \ldots, E(\mathbf{x}|c_n)$, which are learned conditional distributions on underlying latents $c_i$. Latents we consider including position, size, color, gender, hair style, and age, which we refer to as concepts. Figure 2 shows three concepts and their combinations.

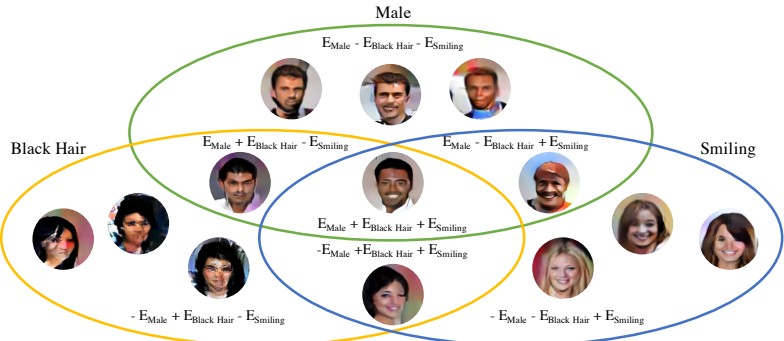

Figure 2: Illustration of concept conjunction and negation. All the images are generated through the conjunction and negation of energy functions. For example, images in the central part is the conjunction of male, black hair, and smiling energy function.

**Concept Conjunction**    In concept conjunction, given separate independent concepts such as a particular gender, hair style, and facial expression, we wish to construct an output with the specified gender, hair style, and facial expression – the combination of each concept. Since the likelihood of an output given a set of specific concepts is equal to the product of the likelihood of each individual concept, we have Equation 4, which is also known as the product of experts (Hinton, 2002)

$$p(x|c_1 \text{ and } c_2, \dots, \text{ and } c_i) = \prod_i p(x|c_i) \propto e^{-\sum_i E(x|c_i)}. \tag{4}$$

We can thus apply Equation 3 to the distribution that is the sum of the energies of each concept to obtain Equation 5 to sample from the joint concept space.

$$\tilde{\mathbf{x}}^k = \tilde{\mathbf{x}}^{k-1} - \frac{\lambda}{2}\nabla_{\mathbf{x}}\sum_i E_\theta(\tilde{\mathbf{x}}^{k-1}|c_i) + \omega^k, \ \omega^k \sim \mathcal{N}(0,\lambda) \tag{5}$$

**Concept Disjunction**    In concept disjunction, given separate concepts such as the color red and the color blue, we wish to construct an output that is either red or blue – either of the given concepts. Thus, we wish to construct a new distribution which has probability mass when any of the chosen concepts are true. A natural choice of such a distribution is the sum of the likelihood of each concept:

$$p(x|c_1 \text{ or } c_2, \dots \text{ or } c_i) \propto \sum_i p(x|c_i). \tag{6}$$

If we assume partition functions for each $p(x|c_i)$ are equal, we can make a further simplification:

$$\sum_i p(x|c_i) \propto \sum_i e^{-E(x|c_i)} = e^{\text{logsumexp}(-E(x|c_i))}, \tag{7}$$

where $\text{logsumexp}(f_1, \dots, f_N) = \log\sum_i \exp(f_i)$. We can thus apply Equation 3 to the distribution that is a negative smooth minimum of the energies of each concept to obtain Equation 8 to sample from the disjunction concept space.

$$\tilde{\mathbf{x}}^k = \tilde{\mathbf{x}}^{k-1} - \frac{\lambda}{2}\nabla_{\mathbf{x}}\text{logsumexp}(-E(x|c_i)) + \omega^k, \ \omega^k \sim \mathcal{N}(0,\lambda) \tag{8}$$

In our experiments, we empirically found the partition function estimates to be similar across concepts (see Appendix A.6), which justifies the simplification in Equation 7.

**Concept Negation**    In concept negation, we wish to generate an output that does not contain the concept. Given a color red, we want an output that is of a different color, such as blue. Thus, we want to construct a distribution that places high likelihood to data that is outside a given concept. One choice is a distribution inversely proportional to the concept. Importantly, negation must be defined with respect to another concept to be useful. The opposite of alive may be dead, but not inanimate. Negation without a data distribution is not integrable and leads to a generation of chaotic textures which, while satisfying absence of a concept, is not desirable. Thus in our experiments with negation we combine it with another concept to ground the negation and obtain an integrable distribution:

$$p(x|\text{not}(c_1), c_2) \propto \frac{p(x|c_2)}{p(x|c_1)^\alpha} \propto e^{\alpha E(x|c_1) - E(x|c_2)} \tag{9}$$

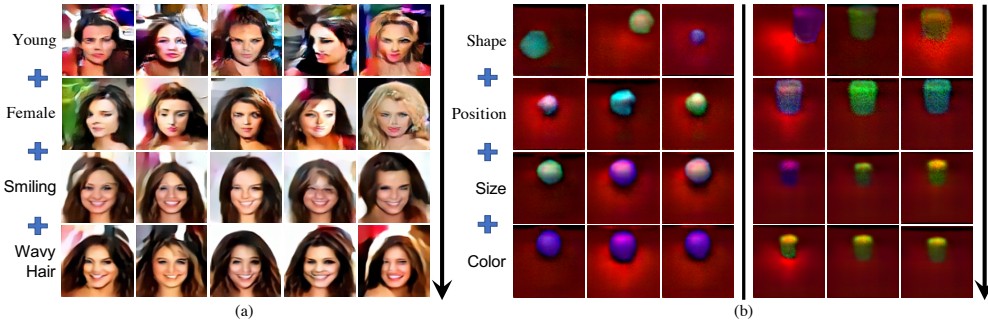

(a)                                                                                    (b)

Figure 3: Combinations of different attributes on CelebA (a) and Mujoco scenes (b) via summation of energies. Each row adds an additional energy function attribute. For example, (a) images on the first row are only conditioned on young while images on the last row are conditioned on young, female, smiling and wavy hair; (b) images on the first column are only conditioned on shape while images on the last column are conditioned on shape, position, size and color. The left of (b) is the generation of a sphere shape and the right is a cylinder.

Where the last proportionality again follows from our assumption that partition functions for $c_1$ and $c_2$ are equal. We found relative smoothing parameter $\alpha$ to be a useful regularizer (when $\alpha = 0$ we arrive at uniform distribution) and always use $\alpha \approx 0.001$ in our experiments. The above equation allows us to apply Langevin dynamics to obtain Equation 10 to sample concept negations.

$$\tilde{\mathbf{x}}^k = \tilde{\mathbf{x}}^{k-1} - \frac{\lambda}{2}\nabla_{\mathbf{x}}(\alpha E(x|c_1) - E(x|c_2)) + \omega^k, \ \omega^k \sim \mathcal{N}(0, \lambda) \tag{10}$$

We note that the combinations of conjunctions, disjunctions and negations allow us to specify more complex logical operators such as implication, but leave exploration of this to future work.

**Concept Inference**    In concept inference, we infer the latent concept parameters through which a given input is generated. Given several example inputs of an underlying concept, we wish to combine the data to make an informed estimation of the underlying concept. Assuming each input is independent of each other, the overall likelihood of the inputs is equivalent to the product of likelihood of each input under a concept and thus is the conjunction of likelihood of each individual data point

$$p(x_1, x_2, \ldots, x_n|c) \propto e^{-\sum_i E(x_i|c)}. \tag{11}$$

We can then obtain maximum a posteriori (MAP) estimates of concept parameters by minimizing the logarithm of the above expression, assuming that partition functions are equal (justified in the appendix)

$$c(x_1, x_2, \ldots, x_n) = \arg\min_c \sum_i E(x_i|c). \tag{12}$$

## 3  EXPERIMENTS

We perform empirical studies to answer the following questions: Can EBMs exhibit concept compositionality, such as concept negation, conjunction, and disjunction, in generating images? Can we take advantage of concept combinations to learn new concepts in a continual manner? Does explicit factor decomposition enable better generalization? Can we perform inference across multiple inputs?

### 3.1  SETUP

We perform experiments on 64x64 different object scenes rendered in Mujoco (Todorov et al., 2012) and the 128x128 CelebA dataset. For scenes rendered in Mujoco, we generate a central object of shape either sphere, cylinder, or box of varying size and height, with some number of (specified) additional background objects. Images are generated with varying lighting and objects.

We use the ImageNet32x32 architecture and ImageNet128x128 architecture from (Du & Mordatch, 2019) with the Swish activation (Ramachandran et al., 2017) on Mujoco and CelebA datasets. Models are trained on Mujoco datasets for up to 1 day on 1 GPU and for 1 day on 8 GPUs for CelebA. More training details and model architecture can be found in and A.3 and A.4.

### 3.2  COMPOSITIONAL GENERATION

We evaluate EBMs on generation images via compositionality operations of the previous section.

Figure 4: (a) Examples of concept disjunction on joint attributes (represented by conjunction of energy) of not smiling+female and smiling+male. EBMs are able to reliably support concept disjunction (generation of either one concept or the other) even when the concept itself is compound. (b) Examples of concept negation on the attributes of smiling female. When negating the female energy in combination with the smiling energy function, we are able to generate photos of males that are smiling.

**Concept Conjunction**    We find that in Figure 3 (a) that EBMs are able to combine independent concepts of age, gender, smile, and wavy hair with each additional attribute allowing more precise generation. Similarly, we find in Figure 3 (b) that EBMs are able to combine independent concepts of shape, position, size, and color together to generate more precise generations.

**Concept Disjunction**    We also find that EBMs are able to combine concepts additively (generate images that are concept A or concept B) as shown in Figure 4 (a). By constructing sampling using logsumexp, EBMs are able to either sample an image that is not smiling female or smiling male, where both not smiling female and smiling male are specified through the conjunction of energies of the two different concepts. This result also shows that concept disjunction can be chained on top of other operators such as concept conjunction.

**Concept Negation**    We further generate concepts that are the opposite of the trained concept in Figure 4 (b), where we find that negating female, in combination with smiling leads to generation of a smiling male. Furthermore, we note that the ability of concept conjunction, disjunction, and negation allows us to flexibly specify any set of pairwise concepts.

**Multiple Object Combination**    Finally, we explore the use of an EBM to model single object-based concepts. To investigate this, we constructed a dataset consisting of a central green cube with size and position annotations, in conjunction with large amount background clutter objects (which are not green), in which we trained a conditional EBM.

Despite the fact that the training dataset does not have any other green cubes, we find that adding two conditional EBMs conditioned on two different position and sizes, allowing us to selectively generate two different cubes in Figure 5. Furthermore, we find that such generation is able to satisfy the constraints of the dataset. For example, when two conditional cubes are too close, the conditionals EBMs are able to default to generating one cube.

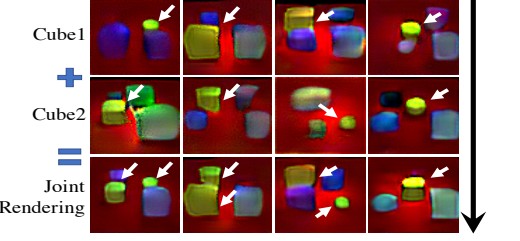
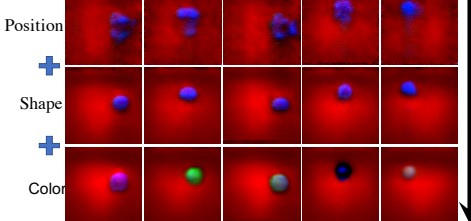

Figure 5: Multi-object compositionality with EBMs. An EBM is trained to generate a green cube of specified size and shape in a scene alongside other objects. At test time, we sample from conjunction of two EBMs conditioned on different position/size attributes (shown in panels cube 1 and cube 2), which generates cubes at both locations. Two cubes are merged into one if they are too close (right column).

Figure 6: Continual learning of concepts. A position EBM is trained on cubes of one color. A shape EBM is then trained on shapes of some fixed color. Finally, a color EBM is trained on shapes of many colors. EBMs continually learn to generate many shape colors at many positions, despite position EBM only being trained on cubes of fixed color, and shape EBM being only trained on shapes of a fixed color.

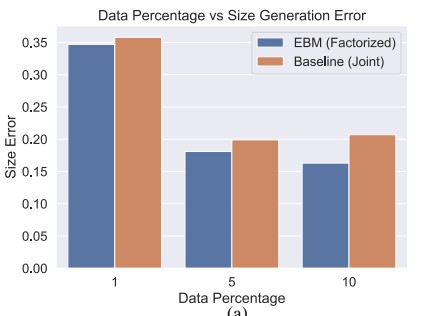
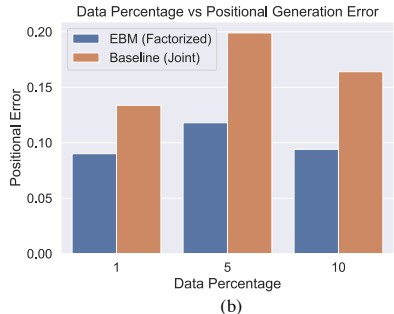

Figure 7: Illustration of generation of size/position concepts as a function of data percentage. By learning a composable representation of underlying concepts, EBMs are able to extrapolate better with less data, and exhibit both lower size and positional error.

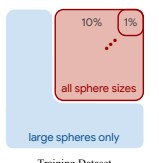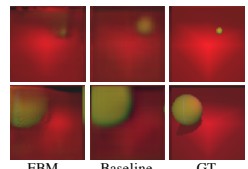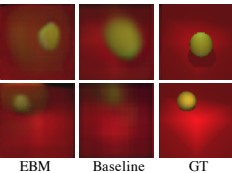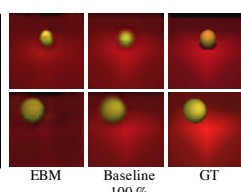

Figure 8: Generated images of novel size and position combinations of EBM and holistic baseline model (right). Training dataset consists of all possible sizes of spheres at (1%, 10%, 100% respectively) of all positions (left).

## 3.3 CONTINUAL LEARNING

An important ability humans are endowed with is the ability to both continually learn new concepts, and to extrapolate existing concepts in combination with previously learned concepts. We evaluate to what extent compositionality in EBMs enables this through the following continual learning process:

1. We first train an EBM for position based generation by training it on a dataset of cubes at various positions of a fixed color.
2. Next we train an EBM for shape based generation, by training the model in combination with the positional model to generate images on a dataset of different shapes (through summation) at different positions, but with the position based EBM fixed.
3. Finally we train an EBM for color based generation, by training the model in combination with both positional and shape models to generate images on a dataset of different shapes at different positions and colors (through summation). Again we fix both position and shape EBMs, and only train the color based generation.

We show in Figure 6 that this allow us to **extrapolate** our learned models for position and shape to generate different position shapes of various colors. The first column of Figure 6 shows the generations of a positional model, while the second column shows the generation of both positional and shape models, and the third column columns shows the generation of position, shape, and color models. Even though the positional model has only seen cubes of a particular color at a particular position, and color model shapes of a particular color, the third column illustrates that the composition of all three models is able to allow the generation of different colored shapes at various positions.

## 3.4 CROSS PRODUCT GENERALIZATION

Humans are endowed with the ability to extrapolate novel concept combinations when only a limited number of combinations were originally observed. For example, despite never having seen a purple person, one might compose what such a person looks like.

We evaluate the extent to which EBMs, which allow us to factorize generation into different concepts, can help us extrapolate. To test this, we construct a dataset of mujoco scene images with spheres of all possible sizes appearing in an increasing percentage of positions starting from top right corner and spheres of only large size appearing in all remaining positions (see left Figure 8 for illustration), *e.g.* 1% means only 1% of positions starting from top right corner with all sphere sizes are used for training. At test time, we evaluate generation of spheres of all sizes at positions not seen at training time. Such a task requires the model to extrapolate the learned position and size concepts in novel combinations.

Table 1: Position prediction error on different test datasets. "Test" has the same data distribution with training set. Other datasets change one or more environmental parameters, e.g. color, size, type, and light, which are unseen in the training set. "Avg" is the average error of "Color", "Light", "Size", and "Type". "Steps"indicates the number of negative sampling steps used to train the EBM. EBMs are able to generalize better. Larger number of negative sampling steps significantly decrease overall EBM error.

| Model | Steps | Color | Light | Size | Type | Avg | Test |
|-------|-------|-------|-------|------|------|-----|------|
| EBM | 80 | 11.172 | 8.458 | 13.201 | 7.107 | 9.985 | 5.582 |
| EBM | 200 | 10.899 | 6.307 | 8.431 | 6.304 | 7.985 | 3.903 |
| EBM | 400 | **4.084** | **4.033** | **6.853** | **3.694** | **4.666** | **2.917** |
| Resnet | - | 20.002 | 5.881 | 10.378 | 6.310 | 10.643 | 3.635 |
| PixelCNN | - | 60.607 | 58.589 | 33.889 | 48.138 | 50.306 | 43.460 |

Figure 9: The influence of multiple observations on EBMs. Multiple images are generated under different lighting conditions and objects. (a) The position prediction error decreases when the number of input images increases independent of negative training steps used to train models. (b) Examples of generated images with varying number of negative sampling steps. Large number of steps leads to more realistic images.

We train two EBMs on this dataset - one conditioned on position latent and trained only on large sizes and another conditioned on size latent and trained at the aforementioned percentage of positions. Conjunction of the two EBMs is fine-tuned on this dataset using gradient descent on Equation 2. We compare this composed model with a baseline holistic model conditioned on both position and size jointly, trained on the same position/size combinations, and optimized directly for Mean Squared Error loss. Both models use the same architecture and number of parameters (described in Appendix).

Qualitatively, Figure 8 illustrates that by learning an independent model for each concept factor, it is possible to combine them to form images from very few combination examples. For reference, both combined and baseline models generate accurate images when given examples of all combinations.

To evaluate generation performance quantitatively, we train a regression model that outputs both the position and size of a generated sphere image. We show errors in regressed size and position from combination and baseline models from ground truth in figure Figure 7. We find that EBMs are able to extrapolate both position and size better than a baseline joint model. We note that both models obtain less positional generation error at 1% data as opposed to 5% or 10% of data. This result is due the make-up of the data – with 1% data, only 1% of the rightmost sphere positions have different size annotations, so failed extrapolation causes models to generate large spheres at the conditioned position. Once there are more different size sphere annotations from data, models either collapse to an existing size or position, leading to a higher error.

### 3.5 COMPOSITIONAL CONCEPT INFERENCE

In addition to generation, EBMs can be used to infer the underlying concept latent factors that gave rise to a particular image. We show this process can also infer compositions of underlying factors.

**Concept Parameter Inference**   To infer underlying concept parameters given an image, we minimize concept energy with respect to these parameters (Equation 12). To evaluate inference accuracy, we generate a mujoco scene image dataset with spheres and cubes at all possible positions under varying lighting and objects. We train an EBM on this dataset using our standard objective and evaluate Mean Absolute Error between inferred object position and ground truth position.

We construct a variety of test datasets that specifically use combinations of concept not seen during training to test generalization of position inference. *"Color"* refers to a test dataset with object colors never before seen in training. *"Light"* is a test dataset with different light sources. *"Size"* is a test dataset with object sizes not seen during training and *"Type"* dataset consists of cylinder images while the training images are only spheres or cubes.

We compare EBMs with a baseline ResNet model (He et al., 2016) (with the same architecture as the EBM) trained directly to regress on true position and a conditional generative baseline PixelCNN

(Oord et al., 2016). Table 1 shows the comparisons of EBMs with different number of Langevin Dynamics sampling steps and the ResNet model. EBMs with larger Langevin sampling steps outperform the ResNet model, generalizing significantly better. Furthermore, larger numbers of sampling steps leads to better generation quality as shown in Figure 9 (b). This suggests that many figures in this paper (which are trained with 40 sampling steps) can likewise see a large boost in the generation quality.

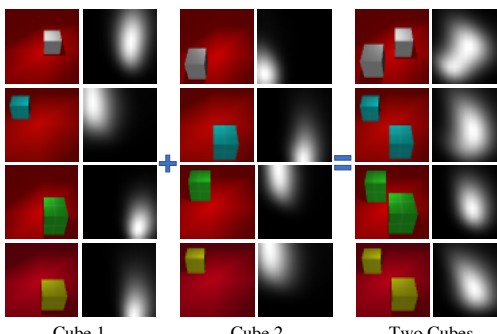

Figure 10: Inference with EBM trained on single cubes and tested on two cubes. In color is input image and in grayscale is energy over object positions. The energies for images of two cubes correctly infer bimodality.

**Inference from Multiple Observations** Another advantage of compositionality in EBMs is the ability to effectively make use of multiple observations when making inferences. We evaluate this ability by measuring the Mean Absolute Error in position when given successively more images with varying of objects and lighting angles. We display results in Figure 9 (a) and find that our model is able to integrate multiple observations to improve prediction performance.

**Inference of Multiple Objects** We also investigate the role of compositionality in generalizing inference to multiple objects. Given EBMs trained on images of a single object, we test on images with multiple objects (not seen in training). We plot energies over possible position values in Figure 10 as well as energy maps over each individual cube. We find that EBMs naturally form bimodal object position distributions that match the summation of distributions for each individual object. We need to specify the number of scene objects for this inference process.

## 4 RELATED WORK

Our work draws on results in energy based models - see (LeCun et al., 2006) for a comprehensive review. A number of methods have been used for inference and sampling in EBMs, from Gibbs Sampling (Hinton et al., 2006), Langevin Dynamics (Du & Mordatch, 2019), Path Integral methods (Du et al., 2019) and learned samplers (Kim & Bengio, 2016; Song & Ou, 2018). In this work, we show that MCMC sampling on EBMs through Langevin Dynamics can be used to compositionally generate realistic images.

Compositionality has been incorporated in representation learning (see (Andreas, 2019) for a summary) and in generative modeling. One approach to compositionality has focused on learning disentangled factors of variation (Higgins et al., 2017; Kulkarni et al., 2015; Vedantam et al., 2018). Such an approach allows the combinatorial specification of outputs, but does not allow the addition of new factors. A different approach to compositionality includes learning various different pixel/segmentation masks for each concept (Greff et al., 2019; Gregor et al., 2015). However such a factorization may have difficulty capturing the global structure of an image, and in many cases different concepts can not be explicitly factored as attention masks.

In contrast, our approach towards compositionality focuses on composing separate learned probability distribution of concepts. Such an approach allows viewing factors of variation as constraints (Mnih & Hinton, 2005). (Hinton, 1999) shows that product of EBMs allows for conjunction of different concepts. In our work we illustrate additional logical compositions and corresponding performance on realistic datasets.

Our work is motivated by the goal of continual lifelong learning - see (Parisi et al., 2018) for a thorough review. Many methods are focused on how to overcome catashtophic forgetting (Kirkpatrick et al., 2017; Li & Hoiem, 2017), but do not support dynamically growing capacity. Progressive growing of the models (Rusu et al., 2016) has been considered, but is implemented at the level of the model architecture, whereas our method is agnostic to the models. Meta and few-shot learning (Reed et al., 2017; Bartunov & Vetrov, 2018) is another approach, but focuses on learning to model images rather than factors.

## 5 CONCLUSION

We have presented work demonstrating the potential of EBMs for both compositional generation and inference and hope to inspire future work in this direction. While we have focused on labeled latents,

we believe an interesting question would be to further study learned unsupervised concepts from unstructured images. We further provide a compositional logic to compose generation with different EBMs, which we note is fully differentiable, allowing further exploration of end to end differentiable training throughout composition.

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

# A  APPENDIX

## A.1  ADDITIONAL COMPOSITIONALITY RESULTS

We present the composition of old, male, smiling, and nonwavy hair trained on CelebA in Figure 11 and composition of old, male, smiling and wavy hair in Figure 12a (as well as a baseline comparison of composition of male/wavy hair in Figure 12b)

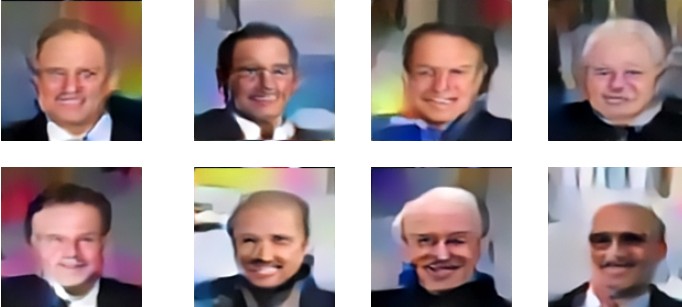

Figure 11: Generated images from the composition of an EBM trained on old, male, smiling and nonwavy hair.

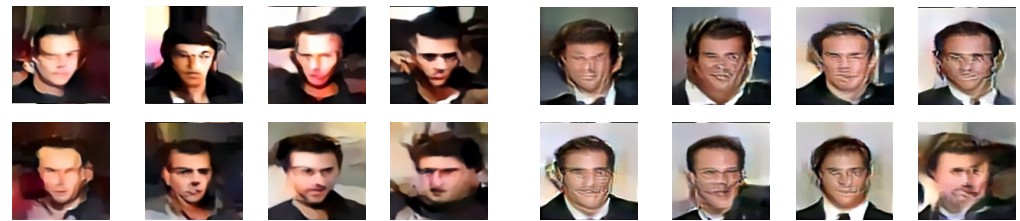

(a) Generated images from the composition of an EBM trained on old, male, smiling and wavy hair.

(b) Generated images from the composition of an EBM trained on wavy and male factors.

Figure 12: Composition of ebms

We also consider combining EBMs from different domains together in Figure 13. We combine a conditional EBM trained on the attribute smiling on CelebA with an EBM trained on Mujoco scenes on different different plane colors. EBMs are trained on separate datasets with separate architectures, but are still able to successfully generate meaningful combinations.

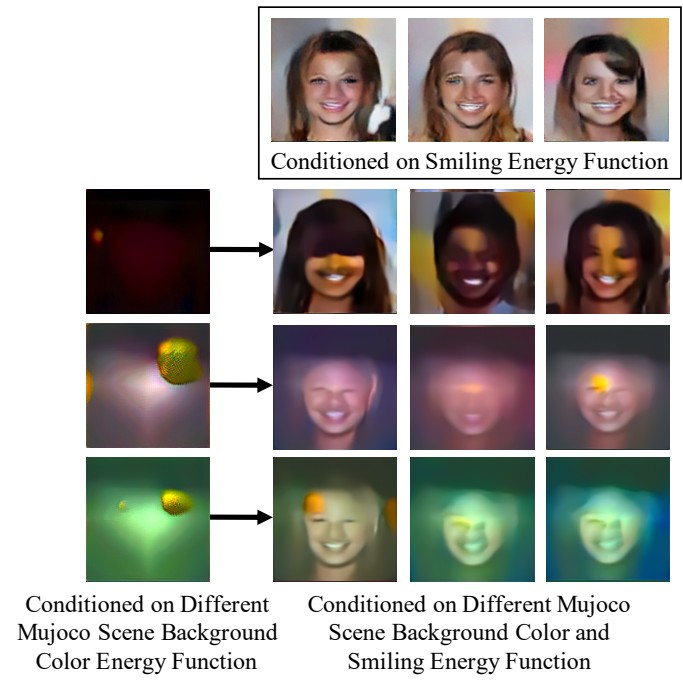

Figure 13: Generated images from EBMs trained on different domains. One EBM is conditioned on the attribute of smiling from the CelebA dataset, while the other EBM is conditioned on the color of the plane from a Mujoco Scenes dataset.

## A.2 DISCUSSION ON OTHER GENERATIVE MODELS

To sample from the conjunction of two seperate probability distribution, MCMC can also be run on the likelihood of other deep likelihood based generative models. However, other schemes besides MCMC sampling, such as autoregressive sampling are not supported, since, there is no natural way to sample from the product of two real distributions without using MCMC.

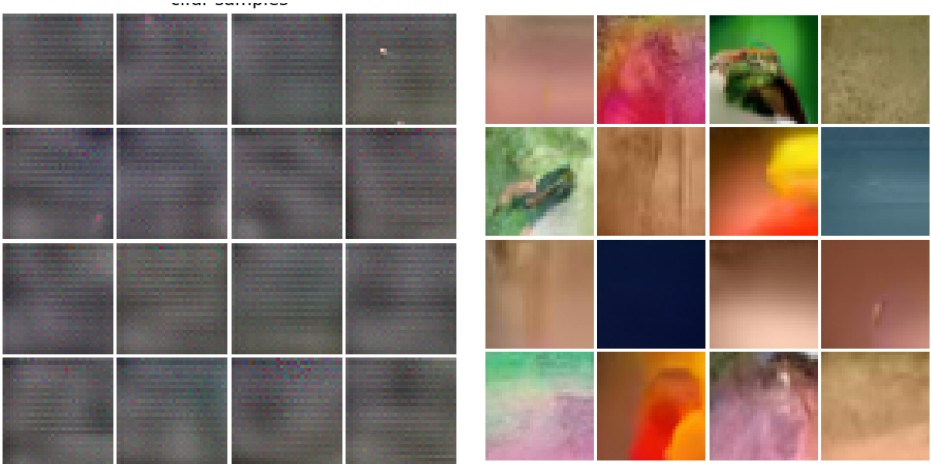

(a) Samples Generated from Langevin Sampling on PixelCNN++ model from (Salimans et al., 2017)

(b) Samples Generated from Autoregressive Sampling on PixelCNN++ model from (Salimans et al., 2017)

Figure 14: Comparison on samples generated from different sampling scenes on PixelCNN++ model from (Salimans et al., 2017). We note that Langevin sampling, while not making realistic samples, generate **higher** likelihood samples than those from autoregressive sampling

We considered Langevin based sampling on the pretrained CIFAR-10 unconditional PixelCNN++ model (Salimans et al., 2017) in Figure 14. While both sampling schemes generate images with similar likelihoods (with Langevin sampling creating higher likelihood samples), we find images generated from Langevin sampling are significantly poorer than those generated from autoregressive sampling. We believe that when using MCMC sampling on generative models, it best to use EBMs since they are trained with MCMC inference, while other models are not trained in such a manner, and may have modes easily found through sampling that are not realistic as noted by (Nalisnick et al., 2018).

## A.3 MODELS

| 3x3 conv2d, 64 |
| --- |
| ResBlock down 64 |
| ResBlock down 128 |
| ResBlock down 128 |
| ResBlock down 256 |
| Global Mean Pooling |
| Dense → 1 |

(a) Mujoco Scenes Model

| Dense → 4096 |
| --- |
| Reshape → 256x4x4 ResBlock up 256 |
| ResBlock up 128 |
| ResBlock up 64 |
| ResBlock up 64 |
| 3x3 conv2d, 3 |

(b) Section 3.4 Baseline Model

| 3x3 conv2d, 64 |
| --- |
| ResBlock down 64 |
| ResBlock down 128 |
| ResBlock down 256 |
| ResBlock down 512 |
| ResBlock down 1024 |
| ResBlock 1024 |
| Global Sum Pooling |
| dense → 1 |

(c) CelebA Model

We detail the EBM architectures used for the Mujoco Scenes images in Figure 15a and for the Celeba 128x128 images in Figure 15c. The baseline model used in comparison for section 3.4 is in Figure 15b

## A.4 TRAINING DETAILS/HYPERPARAMETERS/SOURCE CODE

We include an anonymous link to code used in our experiments in https://drive.google.com/file/d/138w7Oj8rQl_e40_RfZJq2WKWb41NgKn3/view?usp=sharing

When training models on both Mujoco Scenes and CelebA datasets, we use the Adam optimizer with a learning rate 3e-4 with first order moment 0.0 and second order moment 0.999. We use a batch size of 128 to train models, and use a replay buffer of size 50000, with a 5% replacement rate. We apply spectral normalization across models and use a step size of 100 for each Langevin dynamics step. We use 60 steps of Langevin sampling per training iteration for CelebA dataset and 80 steps of Langevin sampling per training iteration for the Mujoco Scenes dataset. We use the Swish activation to train our models (as noted in (Du & Mordatch, 2019)), and find that it greatly stabilizes and speed up training of models.

## A.5 QUALITATIVE COMPARISON TO UNCONDITIONAL MODEL

In this section, we present qualitative samples from an unconditional CelebA in Figure 16, using the same training setup as the conditional models.

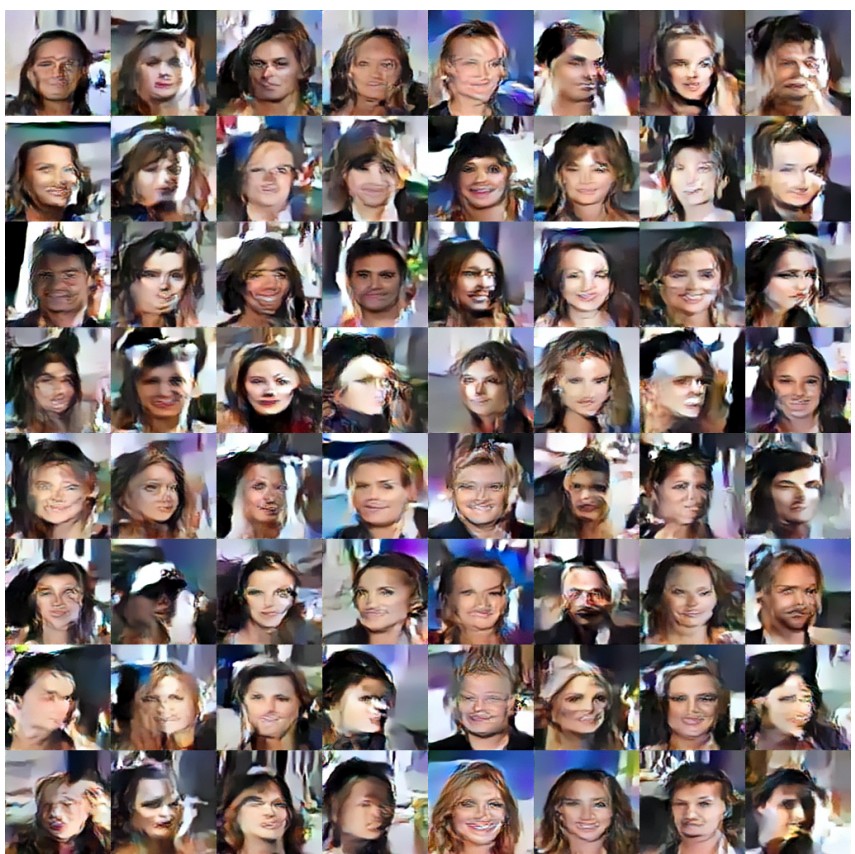

Figure 16: Generated unconditional CelebA images.

We find that conditional generation/unconditional generation in a single model in actually worse than that of compositions of increased numbers of energy models. This shows that compositionality is actually helpful in generation, due to a large number of models helping to refine generation. While these images are worth than those generated by GANs, we believe increased computational power, model size, and number steps

## A.6 Approximating the Magnitude of Partition Function

To approximate the magnitude of the partition function, we can evaluate and plot the histogram energies each model assigns to all data points the model is trained on. Empirically, we found that a combination of L2 normalization and spectral normalization used during training makes it so that EBMs with different architectures and datasets have surprisingly similar histograms as shown in Figure 17.

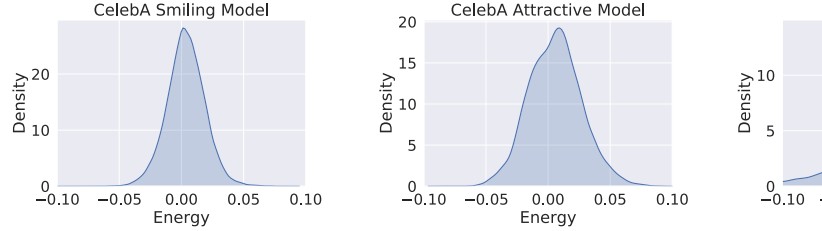

Figure 17: Energy histogram of model trained on CelebA smiling (left), CelebA attractive (middle) and pretrained CIFAR-10 model from (Du & Mordatch, 2019) (right). Are energy histograms are relatively similar.

In Figure 17, we compare the energy histogram of a CelebA model trained on either smiling or attractive histograms as well as the CIFAR-10 model from (Du & Mordatch, 2019). We find that all energy histograms are suprisingly similar, exhibiting minimum and maximum energies between -0.01 and 0.01. This is true even for the CIFAR-10 model which uses a significantly different dataset and

architecture. Given this, we believe that in practice, EBMs exhibit similar partition functions across models at the same temperature. In scenarios in which energy histograms do look different across models on the same dataset, scaling the model by a suitable temperature so that histograms do match can be a useful proxy for equivalent partition functions.

