# OpenReview forum: "Compositional Visual Generation with Energy Based Models"
_ICLR.cc/2020/Conference — Reject_

### Official Review · AnonReviewer1 · 2019-10-23
**Official Blind Review #1**

**Rating:** 6

**Review:**

Update: In light of the quality of the author response, I decide to raise my score to 6: Weak Accept. I encourage the authors to keep improving their writing of section 3.4. The description of the dataset is much clearer than the previous version, but I still think it is worthwhile to use more space for a better description.

There are some other issues with this paper that I recommend the authors to address in the revisions.

- Currently all quantitative experiments are based on conjunctions. It would be nice to have quantitative results for disjunctions and negations as well to make the authors' argument more convincing.

- I would suggest removing the result of PixelCNN in the paper. Of course sampling with Langevin dynamics for PixelCNN is hard. I think a more straightforward way is to do Langevin dynamics for each conditional distribution. The current result of PixelCNN is not done in this way and can be misleading for future work.

That being said, I think the idea proposed by this paper is of novel interest, and even though the execution is not ideal, I feel the novelty itself should warrant an acceptance at ICLR.


============ Original Review ===============

This paper proposes several rules to compose different energy-based models for compositional image generation. The authors propose empirical rules for conjunction, disjunction, and negation, which in combination can be used to derive arbitrary logic expressions. The authors demonstrate that using these composition rules, energy-based models can be used for compositional image generation, continual learning, and compositional concept inference.

I agree with the authors that the composition of energy-based models is a very desirable property and a very important direction to explore, and I believe that the work of this paper can have great potential in the future. However, I feel that the current paper is not very mature, and many questions are left unanswered. Here are some detailed comments:

- What's the justification for the formula of concept disjunction? For disjunction, it makes sense to me to form a mixture of distribution where each mixture has the same weight, ie, \sum_i p(x | c_i). However, \sum_i p(x | c_i) is not proportional to \sum e^{-E(x | c_i)} because of the unknown partition functions. Therefore, when the partition functions of two energy-based models differ by a lot, the disjunction will not be a mixture with equal weights for each component. Instead, it will collapse to a single distribution where all weights are put on the component with the smallest partition function. This does not match the intuition of "disjunction".

- One way of solving the disjunction problem might be abandoning the Langevin dynamics sampling procedure (7). Sampling from a mixture of distribution is doable for arbitrary generative models as long as it is possible to sample from each component of the mixture. Why not just apply Langevin dynamics to each energy-based model in the disjunction and return the samples from each model with the same probability? This procedure cannot be trivially composed with other rules such as conjunction and negation, but it suffices to produce all results in this paper and should be more theoretically sound.

- In Table 1, it seems that energy-based models are more robust in concept inference compared to a ResNet. Why is this the case? I would envisage that the energy-based model is also a ResNet and should have the same inductive bias? Is this a property of generative classifiers? Since the results in Table 1 do not involve composition of energies it is desirable to compare the results with other generative models as well, such as conditional PixelCNN.

- Since more steps of negative sampling can give better samples and generalize better, why not use 400 steps in all experiments throughout the paper?

- In the original OpenAI paper for energy-based models [1], the authors tune the coefficients of the gradient and noise terms in Langevin dynamics for better performance. This is equivalent to doing the ordinary Langevin dynamics for a different energy function re-scaled by some temperature parameter. Did you tune the scales of Langevin dynamics as well? If so, since the energies are scaled by some temperature, the original composition rules will be problematic. Did you take account of this temperature scaling in doing the experiments?

- There is no detailed description of model architecture or hyperparameter in the paper. The author provide no detailed information on how the model was trained, for example, the various hyperparameters in Langevin dynamics and replay buffer. No code is available. There is even no appendix. The paper is very hard to reproduce, which hurts the reliability of the experimental results.

- Writing can be improved.  The description of the dataset in section 3.4 is barely readable. The second paragraph in section 3.4 needs to be paraphrased or significantly expanded. At least there should be some more detailed description of the settings in the appendix.

- At the end of the day, why energy-based models? I can imagine that an autoregressive model can be perfectly used for all tasks described in this paper. It does not even have the problem I mentioned in disjunction, since all densities are normalized. No Langevin dynamics sampling is needed. Since autoregressive models decompose the density using the chain rule, all compositions of the densities can also nicely decompose, and sequential sampling still works.

References:
[1] Du, Yilun, and Igor Mordatch. "Implicit generation and generalization in energy-based models." arXiv preprint arXiv:1903.08689 (2019).

**Experience Assessment:**

I have published one or two papers in this area.

**Review Assessment: Checking Correctness Of Derivations And Theory:**

I carefully checked the derivations and theory.

**Review Assessment: Checking Correctness Of Experiments:**

I assessed the sensibility of the experiments.

**Review Assessment: Thoroughness In Paper Reading:**

I read the paper at least twice and used my best judgement in assessing the paper.

---

> ### Author Response · Authors · 2019-11-11
> **Author Response**
>
> We thank you for your comments.
>
> Regarding our implementation of disjunction and negation, we have updated the paper to make explicit the assumption that partition function is equal across EBMs. To justify our use of this assumption, we added an Appendix A.6 to empirically show that in practice, training EBMs with both L2 normalization and spectral normalization lead to similar energy distributions across data points in dataset regardless of either dataset or models. While calculating true partition function is intractable in our case, the energy histograms point that in practice, the partition function is indeed similar across each EBM.
>
> Further regarding disjunction, implementing concept disjunction as a logsumexp as opposed to discrete sampling has an advantage that the overall selection process is entirely differentiable. While we do not exploit it in this work, differentiability gives us a path towards inferring logical concept combinations in a fully differentiable manner, which could have future applications for unsupervised structure discovery.
>
> For performance in table 1, robustness is indeed a property of generative classifier. However, as demonstrated in [1], EBMs have been shown to be more robust than other generative classifiers, and [2] shows that EBMs are able to match the inference performance of discriminatory models, which most generative classifiers are unable to do. Upon your suggestion, we added a comparison to PixelCNN in Table 1. We believe the robustness in EBMs is not only due to the generative nature but also the negative sampling procedure used to train the models.
>
> Following [1], we trained EBMs with the same temperature coefficient, as we the same coefficient on both noise scale and gradients across all models. To help reproducibility, we have added more implementation details, including network architectures and hyperparameters in the new appendix section. Finally, we provide an anonymous link to our source code at https://drive.google.com/file/d/138w7Oj8rQl_e40_RfZJq2WKWb41NgKn3/view, and will release source code at acceptance.
>
> While we agree that  400 steps of sampling will lead to both better generation and inference, but due to the limitations in computation power, we chose to run a more limited number of steps of sampling to train models (60 or 80 as detailed in the appendix).
>
> We have also updated the writing in the paper and have rewritten section 3.4 and 3.5. We hope the updated sections are more readable and give clarity to our experiment procedures.
>
> We believe that EBMs are the only models that exhibit the compositionality properties well that are demonstrated in the paper. To compose two autoregressive models together in conjunction in a continuous domain, it is necessary to use MCMC sampling. We have added an experiment in Appendix A.1 that shows however, if we try to use the Langevin sampling procedure in autoregressive models, it converges to completely unrealistic images, albeit with the same likelihood as samples sampled in an autoregressive manner (Figure 14). EBMs are explicitly trained with MCMC procedure, making MCMC outcomes on EBMs sensible and allow the models to naturally be combined.
>
> [1] Du, Yilun, and Igor Mordatch. "Implicit generation and generalization in energy-based models." arXiv preprint arXiv:1903.08689 (2019).
> [2] Anonymous. “Your classifier is secretly an energy based model and you should treat it like one” https://openreview.net/forum?id=Hkxzx0NtDB

---

### Official Review · AnonReviewer2 · 2019-10-24
**Official Blind Review #2**

**Rating:** 6

**Review:**

The paper talks about training multiple energy-based models for each concept and then combining them in different ways to construct a composite model. For each composition of concepts, the samples from the composite energy-based model follow the logic of composition.
The main novelty of the paper is to define an energy-based model for each logical composition of the concepts based on the energy-based model of each concept. Regarding the usage of energy-based models, the idea is interesting. However, I am not sure about the existing works using alternative approaches.

My main question: the experiments only combine the energy models trained for the same domain. What does happen if you combine the energy models from different domains, for example, Celebs and color from the objects? What if you combine models with different architectures?

-- The qualitative images show the method is working, but the outputs are not great!!!!

-- I really want to see the outputs for old + male + smiling + heavy hair in contrast to Figure 3?


-- The "concept inference is not clear from the text. The goal is to infer a concept from several given images of the same concept.
It is not clear from the text, but I assume you compute an approximate likelihood (by estimating the partition function with a single sample using Langevin dynamics) of images for each concept and pick the concept with the highest value.
So what do you mean by this: "We can then obtain maximum a posteriori (MAP) estimates of concepts by minimizing the energy of the above expression."

--I had a hard time following the experiments of sections 3.4 and 3.5, mostly because of unclear writing.

--Some sentences are not clear, for example:
"To test this, we construct a sphere dataset consisting of sphere of all sizes at a specified percentage of the rightmost positions and large spheres remaining positions, with size/position annotations."

--Equations 4 and 10 need negation for the energy term to be consistent with Equation 1.
--The citation for Langevin dynamics should be Welling and Teh, 2011.

**Experience Assessment:**

I have published one or two papers in this area.

**Review Assessment: Checking Correctness Of Derivations And Theory:**

I carefully checked the derivations and theory.

**Review Assessment: Checking Correctness Of Experiments:**

I carefully checked the experiments.

**Review Assessment: Thoroughness In Paper Reading:**

I read the paper thoroughly.

---

> ### Author Response · Authors · 2019-11-11
> **Author Response**
>
> We thank you for your comments.
>
> With regards to request for new combinations of concepts, we have added several samples for the combination old + male + smiling + nonwavy hair to the supplement (Figure 11). While our images of compositions on CelebA dataset are not as good as generations from state of the art methods, we show in the supplement that they are similar to image samples from an unconditional EBM (Figure 15). Thus we see that composition of concepts does not lead to reduction in sample quality and that improving baseline quality is orthogonal to our work.
>
> We have further combined EBMs trained on both different domains and architectures  (Figure 12). We train a conditional energy function on different colored plane backgrounds on the Mujoco Scene dataset and composed that with an energy function trained on smiling faces in CelebA. We are happy to add additional combinations of both domains and architectures if requested.
>
> We have rephrased the MAP inference section of text and have further rephrased the experiments in section 3.4 and 3.5 We have updated signs of equations in the paper and added the suggested reference. We hope the updated sections are more readable and give clarity to our experiment procedures.

---

> > ### Comment · AnonReviewer2 · 2019-11-14
> > **Extra experiments**
> >
> > Thanks for the extra experiments. The reason I asked for old + male + smiling + heavy hair is to see the compositional capability of the model to construct images that have rare support in the data. However, the provided old + male + smiling + nonwavy is not very informative in that direction.

---

> > > ### Author Response · Authors · 2019-11-14
> > > **Extra experiments**
> > >
> > > Thanks for the comment. Are there other combinations that would appear to be rare support for you? Unfortunately,  CelebA does not have a heavy hair attribute that we can train an energy model on.  We believe that old + male + smiling + nonwavy  is actually fairly rare support, and the cross dataset compositionality (Figure 12) has zero support. Furthermore, the cube compositionality results (Figure 5) is also zero support (since each model was trained on scenes with 1 cube). Our model performs fairly well in terms of 0 support combinations, so we are happy to run additional combinations if there are any that are more convincing to you.

---

> > > > ### Comment · AnonReviewer2 · 2019-11-14
> > > > **Extra experiments**
> > > >
> > > > How about old + male + smiling + wavy hair?

---

> > > > > ### Author Response · Authors · 2019-11-14
> > > > > **Extra experiments**
> > > > >
> > > > > Sure! We have attached old+male+smiling+wavy hair in Figure 12.

---

> > > > > > ### Comment · AnonReviewer2 · 2019-11-15
> > > > > > **Extra experiments**
> > > > > >
> > > > > > Thanks. This event is exactly what I was worried about.
> > > > > > So looking at Figure 12, it seems that the inference only considers male and wavy hair and ignored the smiling and old concepts.  I guess this happens because the model rarely has seen the combination in data and at the same time it has stronger support for male + wavy hair, so their energy values dominate the sum.

---

> > > > > > > ### Author Response · Authors · 2019-11-15
> > > > > > > **Extra Experiments**
> > > > > > >
> > > > > > > Thanks for your concern. We believe that inference also consider old and smiling energies, but the features are less apparent. To make this more obvious, we have added a baseline figure just combining male and wavy hair attributes.

---

### Official Review · AnonReviewer3 · 2019-10-26
**Official Blind Review #3**

**Rating:** 3

**Review:**

This paper proposes to combine energy functions to realize compositionality. This is interesting, and different from previous methods, which use either an explicit vector of factors that is input to a generator function, or object slots that are blended to form an image.
Specifically, three operators (logical conjunction, disjunction, and negation) are realized and empirically evaluated through combining energy functions in three different ways.
Extrapolating concept combinations, continually learning, and concept inference are also evaluated.

This paper is well motivated, showing compositional generation and inference for images. However, I have some concerns:

1. The experiments on the CelebA dataset are mainly subjective.

2. The equal sign in Eq.(4) should be \proto.

3. The most serious concern is that although empirical results are promising, I have concern about the correctness that Eq.(6) realizes disjunciton, and Eq.(8) realizes negation.

It is sensible that Eq.(4) realizes conjuction, according to the idea of Product-of-Expert. Multiplying several energy-based densities reduces to summation of the energies.

For Eq.(6), the authors ignore the influence of normalizing constants when adding several energy-based densities. The authors seem to assume that the normalizing constants for p(x|c_i) are equal. Justification is needed.

Note that we cannot have :
log [0.6* exp(-E(x|c1)) + 0.4* exp(-E(x|c2)) ] will output c1 with probability 0.6 and c2 probability 0.4.

Fig. 1 seems to illustrate ideas at first sight, but is not so convinced at second thought.

4. No discussion for the \alpha in Eq.(8) for concept negation.

5. The description of the baseline joint model in Section 3.4 is missing.

6. For learning EBMs, the following reference is missed, besides (Kim & Bengio, 2016)
Yunfu Song, Zhijian Ou. Learning Neural Random Fields with Inclusive Auxiliary Generators. arxiv 1806.00271, 2018.

--------update after reading the response-----------
I appreciate the authors' response, but the paper still lacks in sound justification of assuming equal normalizing constants in concept disjunction and concept negation.
The claim that the partition functions are similar across dataset and models is mainly empirical (hardly hold in general). The authors' comment (under Figure 16 in A.6) on scaling the model by a suitable temperature to force histogram match in practice makes their reasoning further complicated.

Convincing quantitative experiments on disjunction and negation lacks. It would be better to focus on conjunction by EBMs, which already can makes a good paper, instead of claiming skeptical disjunction and negation by EBMs. A mixture of distribution is more natural to realize disjunction (Review #1 also comments on this).

Therefore, I tend to keep my original score.

Minor: in the updated paper, "The equal sign in Eq.(4) should be \proto" is not fixed.

**Experience Assessment:**

I have published in this field for several years.

**Review Assessment: Checking Correctness Of Derivations And Theory:**

I carefully checked the derivations and theory.

**Review Assessment: Checking Correctness Of Experiments:**

I assessed the sensibility of the experiments.

**Review Assessment: Thoroughness In Paper Reading:**

I read the paper thoroughly.

---

> ### Author Response · Authors · 2019-11-11
> **Author Response**
>
> We thank you for your comments.
>
> Regarding theoretical concerns over concept disjunction and negation, we have updated the paper to make explicit the assumption that partition function is equal across EBMs. To justify our use of this assumption, we added an Appendix A.6 to empirically show that in practice, training EBMs with both L2 normalization and spectral normalization lead to similar energy distributions across data points in dataset regardless of either dataset or models. While calculating true partition function is intractable in our case, the energy histograms point that in practice, the partition function is indeed similar across each EBM.
>
> We have further clarified the interpretation and use alpha for negation for our model in the revised paper. In practice, we use alpha values close 0.001.
>
> In the appendix, we have added the model description of baseline join model in section 3.4. We further added additional emperical details for training models and have generated an anonymous source code link at https://drive.google.com/file/d/138w7Oj8rQl_e40_RfZJq2WKWb41NgKn3
>
> We have updated our text with the suggested reference and also fixed the suggested typos.

---

### Author Response · Authors · 2019-11-11
**Revision**

In response to reviewer feedback, we have revised the manuscript in the following ways:

-Made explicit the hidden assumption about equality of partition functions in concept disjunction and negation in section 2.2 and provided empirical evidence that this assumption holds in Appendix A.6.
-Revised sections 3.4 and 3.5 to elaborate on experiment descriptions and improve clarity of writing, including an illustration of dataset generation.
-Added appendix with implementation and comparison to compositionality with autoregressive models.
-Added anonymous source code link in A.4 to aid in reproducibility of our work.
-Provided detailed concept inference equations and process at the end of section 2.2.
-Gave intuition for parameter alpha and guidelines for setting in negation subsection of 2.2.
-Added additional compositions of novel concepts of old, male, smiling, and non-wavy hair, as well as novel combination of EBMs trained on different datasets in A.1.

---

### Decision · Program_Chairs · 2019-12-19

**Decision:**

Reject

**Comment:**

This submission proposes an image generation technique for composing concepts by combining their associated distributions.

Strengths:
-The approach is interesting and novel.

Weaknesses:
-Several reviewers were not convinced about the correctness of the formulations for negation and disjunction.
-The experimental validation of the disjunction and negation approaches is insufficient.
-The paper clarity and exposition could be improved. The authors addressed this in the discussion but concerns remain.

Given the weaknesses, AC shares R3’s recommendation to reject.